# Effective Bio-Slime Coating Technique for Concrete Surfaces under Sulfate Attack

**DOI:** 10.3390/ma13071512

**Published:** 2020-03-26

**Authors:** Keun-Hyeok Yang, Hee-Seob Lim, Seung-Jun Kwon

**Affiliations:** 1Department of Plant Architectural Engineering, Kyonggi University, Suwon 16227, Korea; yangkh@kqu.ac.kr; 2Department of Civil Engineering, Hannam University, 70 Hannam-ro, Daedeok-gu, Daejeon 34430, Korea; heesubjm@naver.com

**Keywords:** bio-slime, sulfate attack, chloride attack, service life, multi-layer diffusion, repair

## Abstract

The service life of concretes exposed to sulfate decreases as the concrete body expands due to the formation of gypsum and ettringite. Bacteria-based repair coating layers, which have been studied lately, are aerobic and very effective on the sulfate attack. In this study, bio-slime repair coating layers were fabricated using bacteria, and chloride diffusion experiments were performed. In addition, the service life of concrete under sulfate attack was evaluated using time-dependent diffusivity and a multi-layer technique. Chloride diffusivity was compared with sulfate diffusivity based on literature review, and the results were used to consider the reduction in the diffusion coefficient. In the analysis results, the service life of concrete was evaluated to be 38.5 years without bio-slime coating layer, but it was increased to 41.5–54.3 years using it. In addition, when the thickness of the bio-slime coating layer is 2.0 mm, the service life can be increased by 1.31–2.15 times if the sulfate diffusion coefficient of the layer is controlled at a level of 0.1 ~ 0.3 × 10^−12^ m^2^/s. Eco-friendly and aerobic bio-slime coating layers are expected to effectively resist sulfate under appropriate construction conditions.

## 1. Introduction

Concrete structures are used in various environments due to their economic efficiency and durability. Reinforced concrete (RC) structures are the combination of reinforcement, such as reinforcing steel and concrete. In recent years, studies have been concentrated on the corrosion of steel reinforcement due to salt damage and carbonation [1,2]. Most of these studies are focused on the corrosion of buried reinforcement or the corrosion initiation time. Concrete, however, can be degraded through the expansion of the concrete body or local cracks, as well as the corrosion of the internal steel reinforcement [3,4].

A representative deterioration phenomenon that affects the concrete body is the sulfate attack. In the deterioration of concrete by sulfate ions, they penetrate into concrete and react with calcium hydroxide, forming gypsum, or with monosulfates or C_3_AH_6_ (tricalcium aluminate hexahydrate—a C_3_A hydrate), forming ettringite (C_3_A·3CaSO_4_·32H_2_O), which causes the expansion of concrete. The generated ettringite causes cracks at the beginning and significantly affects safety through the expansion of the concrete body [5,6].

In general, sulfate control is conducted by controlling the penetration of sulfates and inhibiting the generation of gypsum and ettringite through the use of admixtures. Many studies have been conducted to improve resistance to deterioration caused by the penetration of sulfate ions, but few of them evaluated the service life through quantitative modeling. Studies on single and combined deterioration have been conducted, but they are focused on the evaluation of deterioration using experimental results. Until recently, the evaluation of deterioration or service life under sulfate attack was dominated by the evaluation of the strength reduction [7,8]. In addition, the material modeling levels of many studies were limited to a correlation with the amount of C_3_A and the expansion volume [6,8].

Sulfate deterioration modeling is difficult because it is complex to simulate deterioration caused by the spalling of external concrete over time; it is also difficult to quantify the reaction of internal hydrates and sulfate ions. The Atkinson model used by ACI (American Concrete Institute) as well as the previously researched models only perform modeling of the deterioration depth and cannot consider the effects after cracking or surface spalling [9,10].

The deterioration of sewage facilities, which are lifelines, has become a problem, and ground subsidence caused by the deterioration of internal concrete and the leakage of joints has been reported [11,12]. For the internal erosion of concrete or pipe lines, bacteria-based self-healing materials can be excellent alternatives that can extend the service life of sewage pipes. Many studies have rapidly advanced aerobic bacteria-based concrete development technologies, and some technologies have reached the commercialization stage [13,14,15,16]. Of course, the residual viability of bacteria and the limitations of self-healing require consistent research, but bacteria-based self-healing materials can be used as excellent repair materials because they can complement such shortcomings as peeling-off, which is commonly found in organic repair materials [17,18]. The protective coating technology using fixed bacteria is expected to have a sustainable resistance to harsh environment that gradually degrades concrete performance, unlike conventional coating materials like epoxy that may cause delamination and peel-off [19].

The coating mortar with fixed bacteria was prepared using expanded vermiculite for a carrying bacteria. The glycocalyx was applied to bacteria in culture, which was used as a barrier to protect from the harmful ion attack outside [13].

In this study, bio-slime was prepared using bacteria to effectively control the penetration of sulfate ions, which are a major deterioration factor, and the service life under sulfate attack was evaluated considering the thickness of the coating layer. The time-dependent ion diffusion technique was used for the service life evaluation, and the results were compared with those of the existing deterioration depth model due to sulfate.

## 2. Preparation of Bio-Slime Coating Layers

Expanded vermiculite (EV) and super absorbent polymer (SAP) were used as immobilization materials to provide the living pores of bacteria and to create a neutral (less than pH 10) growth environment. The particle size of EV ranged from 0.25 to 0.36 mm while that of SAP ranged from 0.08 to 0.20 mm. Their densities were 0.25 and 0.70 g/cm^3^, respectively. These materials had high moisture-retaining capacities and neutral pH. In particular, EV could effectively absorb cations (e.g., Ca^2+^ and Mg^2+^) required for the growth of microorganisms (Rhodobacter capsulatus) because it had an excellent cation exchange capacity. As for bacteria, Rhodobacter capsulatus cultured at a concentration of 109 cel/mL was used, and the pH of the culture medium was slightly acid (pH = 6.8) to improve the highly alkaline environment.

Considering the durability improvement and eco-friendliness of the bacterial glycocalyx coating material, ordinary Portland cement (OPC, S company, Sejong, Korea), ground granulated blast furnace slag (GGBS) and fly ash (FA) were used as binders. In the case of aggregate, silica sands with particle sizes of 0.05–0.17 mm, 0.17–0.25 mm, and 0.25–0.70 mm were mixed at the same mass ratio and used. The porous materials for the immobilization of bacteria were used replacing 30% of the aggregate volume during mixing.

Table 1 shows the formulation of the mortar used as a coating material. Table 2 presents the mixing information of the bio-slime coating material used. Figure 1 shows the material preparation and the overview of the bio-slime coating layer. In order to promote sustainable and protective effect to the coating mortar, the glycocalyx was used as a coating materials. For the cultivation of bacteria, glycocalyx consists of polymer skin capsules and a slime layer. The composition ratio of glycocalyx depends on the characteristics of the bacterial strain and constituent of the cultivate media. In addition, the strength characteristics, sulfate resistance, and preparation of the bio-slime structure can be found in the existing literature [13].

## 3. Sulfate Penetration Analysis Considering Bio-Slime Coating Layer

### 3.1. Analysis without Coating Layer

The service life model for sulfate deterioration considered the model developed by Atkinson and Hearne [9,10]. This model was also developed as a software program, and it shows the deterioration depth over time considering the moisture inflow from the ground [20]. Equation (1) is the basic equation developed by Atkinson and Hearne. This model basically assumes that the formation of expandable ettringite inside concrete causes harmful expansion and cracking and assumes that a failure occurs while the deterioration side ($X_{spall}$) thickly peels off from the concrete surface when the deformation caused by the increased volume of ettringite exceeds the fracture energy of concrete.
(1)$$R=\frac{Z_{p}}{t}=\frac{EB^{2}c_{0}D_{i}x\Phi_{Al_{2}O_{3}}}{0.10196\alpha \gamma \left(1-\nu \right)}$$
where, *R* is the deterioration rate of concrete by sulfate ions (m/s), *Zp* is the predicted sulfate penetration depth (m), *t* is the time (s), *c_0_* is the external sulfate concentration (mol/m^3^), *Di* is the sulfate diffusion coefficient in concrete (m^2^/s), *E* is the elastic modulus (kgf/m^2^), *α* is the roughness coefficient of the area where performance deterioration occurs (1.0), *B* is the stress of 1 mol of sulfate that reacts in 1 m^3^ of concrete (=1.8 × 10^−6^ m^3^/mol), *γ* is the energy required for the concrete surface fracture (kgf/m), *ν* is Poisson’s ratio, *x* is the cement content of the target structure (kg/m³), and *ΦAl_2_O_3_* is the aluminum oxide content of the target structure. As seen in Equation (1), the deterioration depth linearly increases with the increase in the diffusion coefficient under constant mix proportions. Accordingly, the service life also shows a linear reduction.

### 3.2. Analysis Considering Coating Layer

When the concrete surface is deteriorated or reinforced, diffusivity from the surface is increased or decreased. In this study, a repaired case, i.e. a case where the surface has a low diffusion coefficient, was assumed. As there is no modeling for the diffusion coefficient of sulfate ions, the diffusion theory of chloride ions was used. In a research that considers the diffusion theory of a multi-layer structure and the time-dependent diffusion coefficient, the flow of ions based on Fick’s second law can be expressed as Equation (2) [21]. In the previous model [22], the time effect was not considered, so that the developed Equation (2) with time effect was used. The diffusion coefficient was controlled with time exponent *m* and specific period (*t*_c_).
(2)$$\left\{\begin{array}{l} C_{1}\left(x,t\right)=C_{S}\sum_{N=0}^{\infty }\alpha^{n}\left[erfc\left\{\frac{2ne+x}{2\sqrt[{}]{\frac{D_{1}}{1-m}{\left(\frac{t_{0}}{t}\right)}^{m}t}}\right\}-\alpha \cdot erfc\frac{\left(2n+2\right)e-x}{2\sqrt{\frac{D_{1}}{1-m}{\left(\frac{t_{0}}{t}\right)}^{m}t}}\right]\\ C_{2}=\frac{2kC_{S}}{k+1}\sum_{n=0}^{\infty }\begin{array}{l} \alpha^{n}erfc\left[\frac{\left(2n+1\right)e+k\left(x-e\right)}{2\sqrt{\frac{D_{1}}{1-m}{\left(\frac{t_{0}}{t}\right)}^{m}t}}\right] \end{array}\\ (t<t_{c}) \end{array}\right\}\;\left\{\begin{array}{l} C_{1}\left(x,t\right)=C_{S}\sum_{N=0}^{\infty }\alpha^{n}\left[\begin{array}{l} erfc\left\{\frac{2ne+x}{2\sqrt[{}]{\frac{D_{2}}{1-m}{\left(\frac{t_{0}}{t_{c}}\right)}^{m}\left[1-m+m\left(\frac{t_{c}}{t}\right)\right]t}}\right\}\\ -\alpha \cdot erfc\frac{\left(2n+2\right)e-x}{2\sqrt{\frac{D_{2}}{1-m}{\left(\frac{t_{0}}{t_{c}}\right)}^{m}\left[1-m+m\left(\frac{t_{c}}{t}\right)\right]t}} \end{array}\right]\\ C_{2}=\frac{2kC_{S}}{k+1}\sum_{n=0}^{\infty }\begin{array}{l} \alpha^{n}erfc\left[\frac{\left(2n+1\right)e+k\left(x-e\right)}{2\sqrt{\frac{D_{1}}{1-m}{\left(\frac{t_{0}}{t_{c}}\right)}^{m}\left[1-m+m\left(\frac{t_{c}}{t}\right)\right]t}}\right] \end{array}\\ (t\geq t_{c}) \end{array}\right\}\;$$
where, *C_1_* and *C_2_* are the chloride concentrations of the concrete surface and body (kg/m³), respectively; *D_1_* and *D_2_* are the diffusion coefficients of the concrete surface and body (m²/s), respectively; *k* is (D_1_/ D_2_)^1/2^; *α* is (1 − *k*)/(1 + *k*); and *e* is the thickness of the surface reinforced through surface repair. The diffusion coefficient of the surface, *D_1_*, was assumed to be smaller than the internal diffusion coefficient, *D_2_*, to simulate the diffusion coefficient of the deteriorated concrete surface. In addition, *m* and *t_c_* were assumed to be the time-dependent index of the diffusion coefficient and the time when the diffusion coefficient stabilizes (30 years), respectively.

## 4. Evaluation of the Service Life of a Concrete Structure Considering Bio-Slime

### 4.1. Derivation of the Bio-Slime Chloride Diffusion Coefficient

To evaluate the penetration of sulfate ions, the diffusion coefficient of chloride ions was indirectly derived and considered for analysis. It is very difficult to experimentally or analytically implement the diffusion coefficient of sulfate ions because it is difficult to directly implement the phase equilibrium and mobility of ions [3]. Previous studies showed that the ratio of sulfate to chloride ions is 1.06/2.06 under distilled water condition, indicating that sulfate ions are approximately 50% of chloride ions. They also experimentally derived the penetration depths of chloride ions and sulfate and found that the ratio of the square root of chloride ion diffusion coefficient and sulfate ion diffusion coefficient has a linear relationship with the penetration depth. The regression analysis of this relationship revealed that the sulfate ion diffusion coefficient is approximately 38.5% of the chloride ion diffusion coefficient, and this tendency is not significantly different from the 50% level, which is the diffusion coefficient ratio in aqueous solution [3].

In this study, the sulfate ion diffusion coefficient was assumed to be 40% of the chloride diffusion coefficient for analysis. The accelerated chloride diffusion coefficient was derived using the NTBULD 492 method, which is the non-steady-state diffusion coefficient. In the NTBUILD 492 method, 10% NaCl solution is applied for cathode, and 0.3 N NaOH solution is adapted for anode to accelerate penetration of chloride ion. Based on initial current and the criteria specified in the method, applied voltage and test period is determined. After the voltage is applied in each test period, silver nitrate solution (0.1 M, AgNO_3_) is utilized as indicator. The diffusion coefficient from the test was calculated through Equations (3) and (4).
(3)$$D_{rcpt}=\frac{RTL}{zF\left(U-2\right)}\times \frac{X_{d}-\alpha \sqrt{X_{d}}}{t}$$
(4)$$\alpha =2\sqrt{\frac{RT}{zFE}}\times erf^{-1}\times \left(1-\frac{2C_{d}}{C_{0}}\right)$$
where $D_{rcpt}$ and R are chloride diffusion coefficient (m^2^/s) and universal gas constant (8.314 J/mol·K), respectively. T denotes absolute temperature (K) and L means thickness of specimen (m). z and F are ionic valence of 1.0 and Faraday constant (96,500 J/V·mol). U means applied potential (V), t denotes test duration time (s). The chloride concentrations of $C_{d}$ and $C_{0}$ are the that at which the color changes and that in the cathode solution (mol/L), respectively. Figure 2 and Figure 3 show the photo of the diffusion test and the results, respectively. A total of six slime coating types were considered, and excellent chloride diffusion coefficients of 0.71–3.39 × 10^−12^ m^2^/s were derived.

### 4.2. Target Structure and Mix for Analysis

The target structure was RC box structure exposed to a high concentration of sulfate, and the mix proportions were assumed to have 30% of GGBFS replacement ratio and the design strength of 35 MPa, which was conventionally used for resisting chemical attack. Figure 4 shows the overview of the target structure. Table 3 show the analysis conditions for sulfate diffusion. In the analysis conditions, the diffusion coefficient of sulfate ions was assumed to be 40% of that of chloride ions, as discussed in Section 3.1. In addition, the sulfate concentration was assumed to be 5,000 ppm, which is considered a medium value in the 1500–10000 ppm range which is the third worst condition of the ACI 318 standard [23,24]. It is very difficult to set the critical sulfate concentration because there is no clear criterion. The condition at which expansion by ettringite begins during the penetration of Na_2_SO_4_ is usually known as approximately 0.12% [25], but 0.18%, which is 1.5 times higher, was considered under the assumption that the surface was continuously deteriorated.

In South Korea, the sulfate concentration (hydrogen sulfide) inside sewage pipes is less than 10 ppm, which is a very low level, and there is the influence of drainage, such as rainfall. Thus, very low deterioration depths have been reported [26].

### 4.3. Analysis of the Service Life under Different Conditions

(1)When the surface is not protected:

For the unprotected surface, analysis was conducted using two methods. When the Atkinson model was used, the input constants shown in Table 4 were applied. In addition, the service life according to the sulfate diffusion coefficient and that according to the external sulfate concentration are shown in Figure 5.

When the cover thickness (60 mm) and external sulfate concentration (5000 ppm) of the sewage pipe, which was the target structure, were considered, its service life was evaluated to be 38.3 years when surface repair was not performed.

Figure 6 shows the sulfate behavior performed using the diffusion coefficient and Fick’s second law. A software that applies the finite difference method was used. When 30% slag was used replacing cement, the diffusion coefficient (D_ref_) was found to be 7.943 × 10^−12^ m^2^/s and the time-dependent index (m) was 0.314 at the reference age. The service life for the 60 mm concrete cover thickness was evaluated to be 49.1 years.

(2)When the surface is protected:

—Service life according to the slime thickness.

For the analysis of the surface protected by bio-slime, Equation (2) was used. When there was no surface repair, the service life was evaluated to be 38.3 years. As the thickness of the coating layer with a diffusion coefficient of 0.6 × 10^−12^ m^2^/s increased to 1.0–5.0 mm, the service life increased from 38.3 to 54.3 years.

Figure 7 shows the service life behavior due to the coating layer change, and Figure 8 shows the sulfate diffusion behavior distribution according to the coating layer thickness after 50 years of sulfate exposure.

—Service life according to the external sulfate concentration.

After fixing the slime thickness at 2.0 mm and the diffusion coefficient of slime at 0.5 × 10^−12^ m^2^/s, the service life was analyzed while the external sulfate ion concentration was increased from 0.3% to 0.5%. With the increase in the concentration of sulfate, the penetration of harmful ions to the inside was further accelerated. The sulfate concentration of the actual sewage pipe, however, was very low (100–200 ppm). As the sulfate concentration increased from 3000 to 7000 ppm, the service life significantly decreased from 138 to 29 years, as shown in Figure 9. In addition, Figure 10 shows the sulfate distribution behavior in each sulfate exposure environment after 30 years.

—Changes in service life due to changes in the diffusivity of the slime layer.

Actually, the sulfate diffusion coefficient of the repair slim layer has the sulfate diffusion characteristics of 0.27~1.31 × 10^−12^ m^2^/s depending on the bioactive properties used. In this study, the service life was evaluated while the diffusion coefficient of the repair slime layer was increased from 0.1 × 10^−12^ to 1.2 × 10^−12^ m^2^/s. Figure 11 and Figure 12 show the service life according to the diffusion coefficient of the slime layer and the distribution of sulfate ions after 40 years.

The analysis results showed that the service life was 83 years when the sulfate diffusion coefficient of the coating layer was 0.1 × 10^−12^ m^2^/s, but it decreased to 50.5, 43, 40.5, and 39.5 years as the diffusion coefficient increased to 0.3, 0.6, 0.9, and 1.2 × 10^−12^ m^2^/s, respectively.

## 5. Conclusions

In this study, the service life of a concrete sewage treatment structure was evaluated using the experimental values of the bio-slime coating layer and the diffusion model that considered multi-layers. The derived results are as follows:(1)The literature survey revealed that the chloride and sulfate diffusion coefficients are proportional to the square root of the molar ratio. The experimental values and the previously proposed values indicated that the ratio of the sulfate diffusion coefficient to the chloride diffusion coefficient ranges from 0.38 to 0.50. In this study, the service life was evaluated using this relationship;(2)When the Atkinson model was used for the sewage culvert box, its service life was evaluated to be 38.5 years under the conditions of a 60 mm cover thickness and a 5000 ppm sulfate concentration. When the critical sulfate concentration was assumed to be 0.18%, the service life by the diffusion law was evaluated to be 49.1 years;(3)The sulfate diffusion coefficient of the slim coating layer that considered the reduced diffusion ratio ranged from 0.27 × 10^−12^ to 1.31 × 10^−12^ m^2^/s, resulting in the excellent diffusion reduction. When there was no coating layer, the service life was evaluated to be 38.5 years under the conditions that considered concrete properties (30% slag substitution, water to binder ratio of 0.4, and cover thickness of 60 mm). Simple bio-slime coating increased the service life to 41.5–54.3 years. In addition, although the thickness of the slime coating layer was 2.0 mm, the service life could be significantly increased to 50.5–83 years if the sulfate diffusion coefficient of the coating layer could be controlled between 0.1 × 10^−12^ and 0.3 × 10^−12^ m^2^/s;(4)The diffusivity of the bio-slime coating materials (EV-immob-polymer) derived by the experiment is approximately 0.3 × 10^−12^ m^2^/s. As bio-slime coating materials are highly resistant to wetland and sulfate exposure conditions, they are expected to be very effective in extending the service life of the existing concrete structures by reducing deterioration.

## Figures and Tables

**Figure 1 materials-13-01512-f001:**
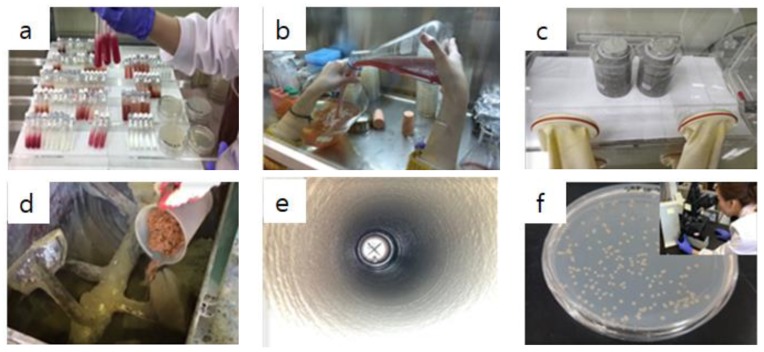
Procedures for producing bio-slime with bacteria and coating for sewage line: (**a**) bacteria lsolation and screening; (**b**) bacteria culture: 10^9^ cell/mL. (pH = 6.8, anaerobic environment); (**c**) bacteria immobilization; (**d**) coating material formulation; (**e**) coating material construction by lining method; (**f**) confirm bacterial growth (recultured and microscopic).

**Figure 2 materials-13-01512-f002:**
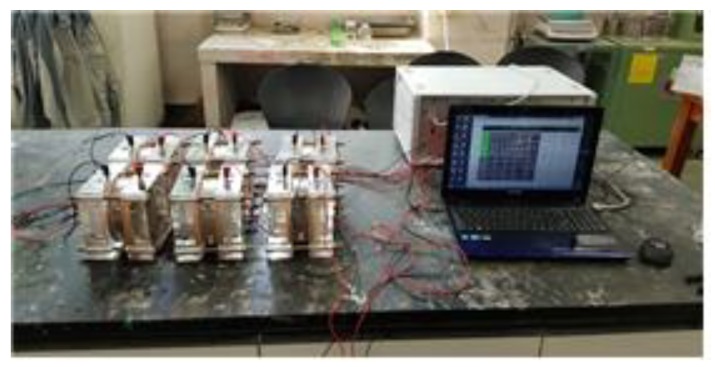
Photo of accelerated chloride diffusion test.

**Figure 3 materials-13-01512-f003:**
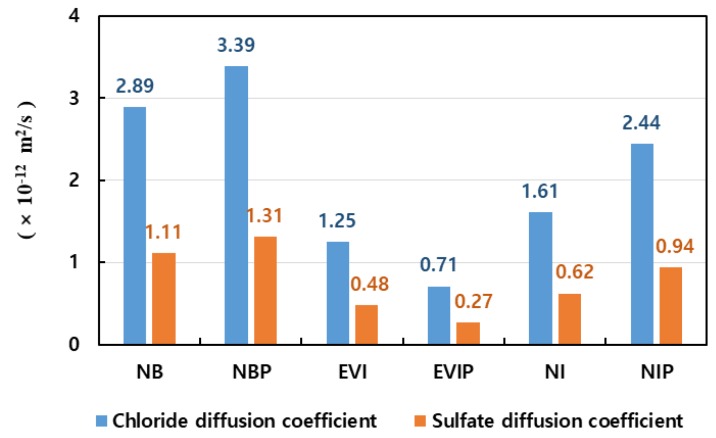
Results from chloride diffusion coefficient in various slime coating conditions. NB: Non-Bacteria; NBP: Non-Bacteria-Polymer; EVI: EV-Immob; EVIP: EV-Immob-Polymer; NI: Non-Immob; NIP: Non-Immob-Polymer.

**Figure 4 materials-13-01512-f004:**
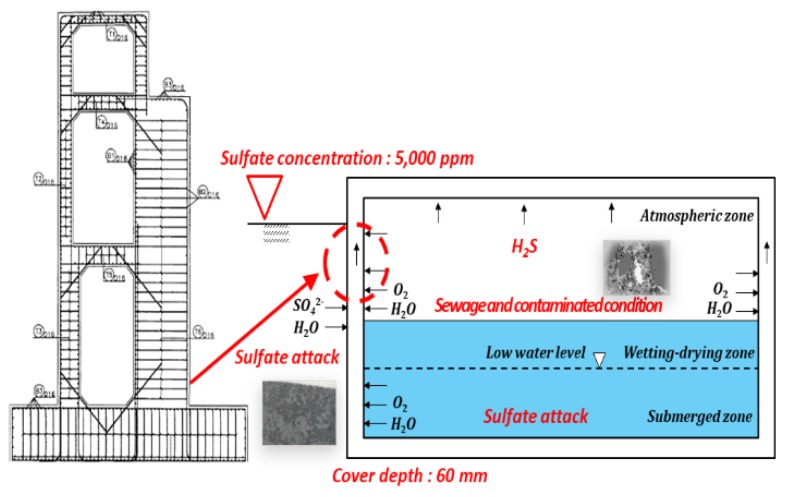
Schematic diagram for the RC structure exposed to sulfate attack.

**Figure 5 materials-13-01512-f005:**
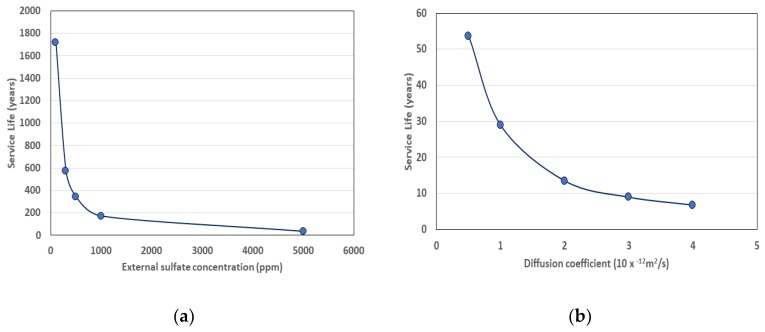
(**a**) Service life variation with exterior concentration; (**b**) service life with diffusion coefficient of sulfate ion.

**Figure 6 materials-13-01512-f006:**
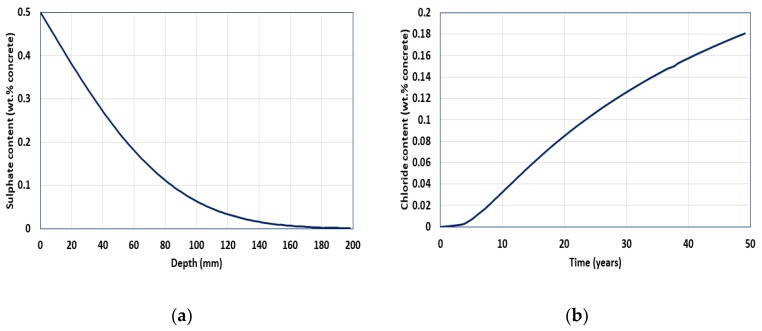
(**a**) Sulfate ion profile with cover depth after 49.1 year; (**b**) increasing sulfate ion at steel location with time.

**Figure 7 materials-13-01512-f007:**
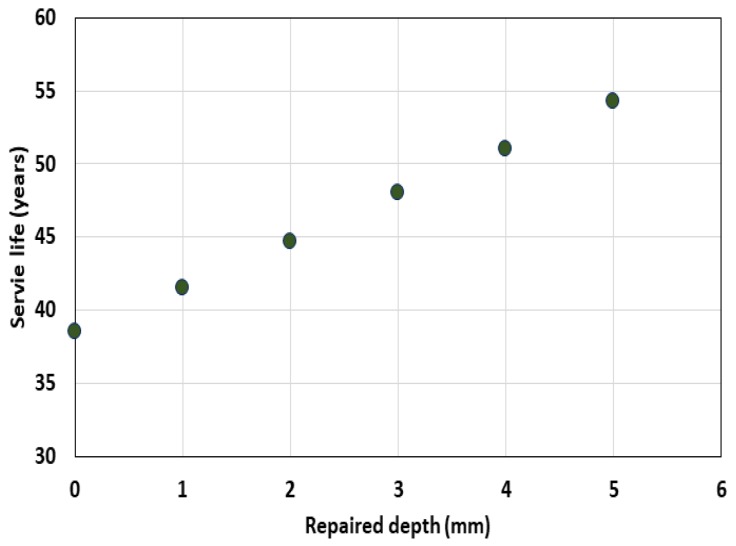
Service life variation with coating thickness.

**Figure 8 materials-13-01512-f008:**
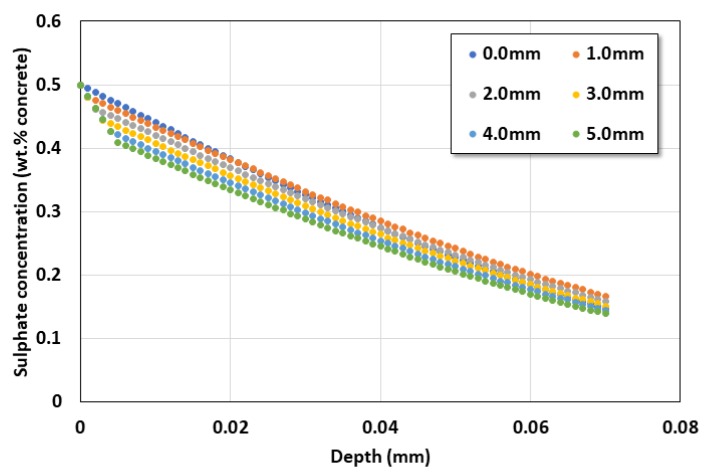
Sulfate ion profile after 50 years.

**Figure 9 materials-13-01512-f009:**
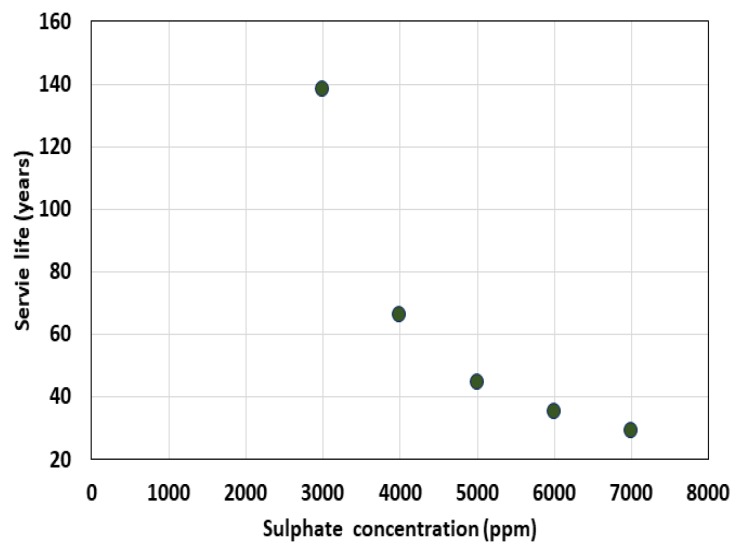
Service life variation with exterior conditions.

**Figure 10 materials-13-01512-f010:**
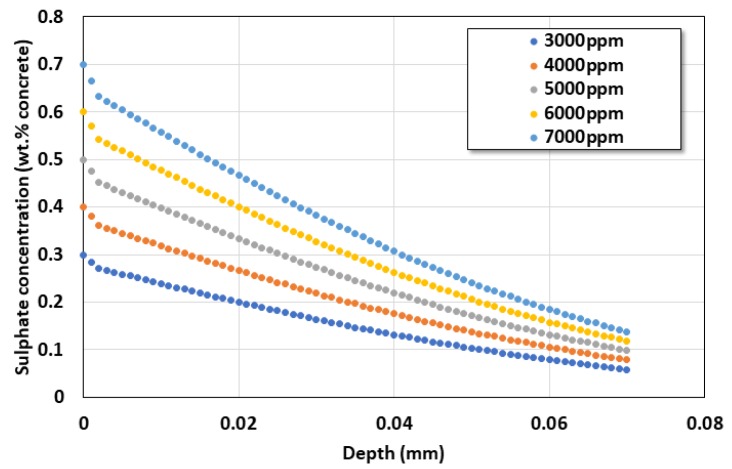
Sulfate ion profile after 30 years.

**Figure 11 materials-13-01512-f011:**
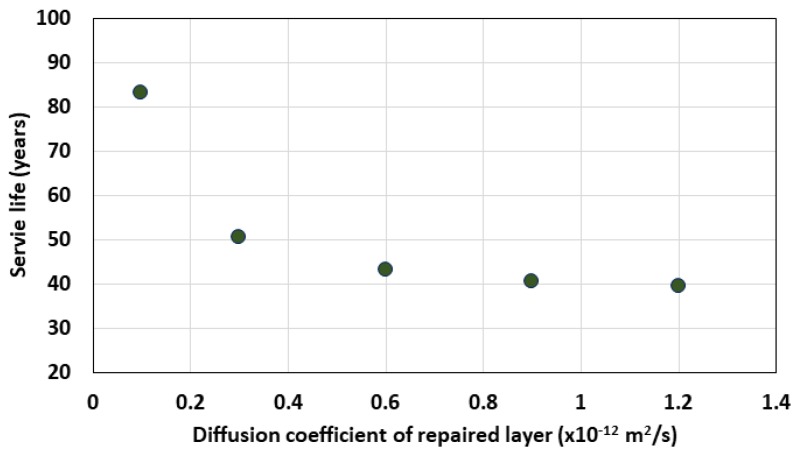
Service life variation with diffusion coefficient in slime coating.

**Figure 12 materials-13-01512-f012:**
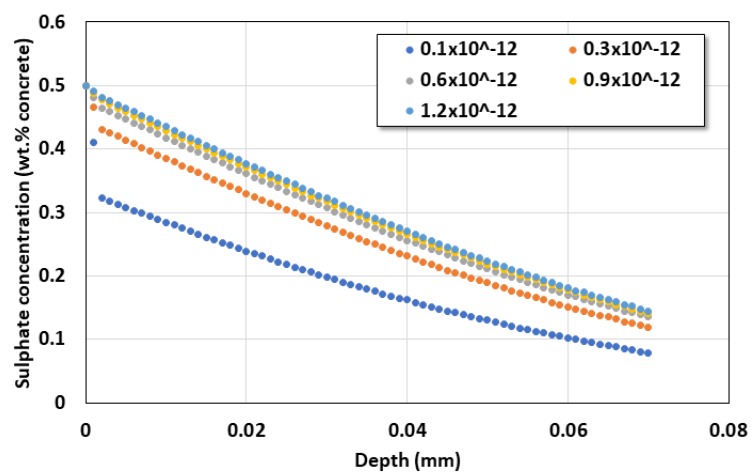
Sulfate ion profile after 40 years.

**Table 1 materials-13-01512-t001:** Mix proportions for mortar.

W/B	S/B	Unit Weight (kg/m^3^)						
		Water	OPC	FA	GGBS	Silica Sand Size (mm)		
						0.05~0.17	0.17~0.25	0.25~0.7
35	2	135.9	158.5	90.6	203.8	196.3	196.3	196.3

**Table 2 materials-13-01512-t002:** Mix proportions for bio-slime coating material.

Sample	Strain	Immobilization Material Type	Immobilization Material Substitution Ratio (%)
Non-bacteria	−	−	−
Non-immobilized bacteria	Rhodobacter capsulatus	−	−
Expanded vermiculite (EV) immobilized bacteria		EV	30
Super absorbent polymer (SAP) immobilized bacteria		SAP	30

**Table 3 materials-13-01512-t003:** Analysis conditions for sulfate diffusion.

Item	Value
Exterior sulfate concentration (%)	5000 ppm
Diffusion coefficient in concrete	7.943 × 10^−12^ m^2^/s
Critical sulfate concentration	0.18% (1800 ppm)
Diffusion coefficient in slime-coating	0.60 × 10^−12^ m^2^/s
Thickness of slime coating layer	1.0~5.0 mm

**Table 4 materials-13-01512-t004:** Input constants for Atkinson model.

Input variable	Unit	Input value	Ground
c0	mol/m³	52.067	5000 ppm sulfate concentration was converted into the molar weight (96.06 g/mol)
Di	m²/s	0.7 × 10^−12^	Mean value of typical sulfate diffusion coefficients ((0−4) × 10^−12^ m^2^/s) based on the results of previous studies
E	Pa	27,800 × 10^6^	Typical elastic modulus of concrete
ν	−	0.177	Typical Poisson’s ratio
x	kg/m^3^	289.1	The amount of cement used in the mix design was used (slag substitution rate: 30%).
ΦAl_2_O_3_	mol/m³	0.05	The mean value of cement manufacturers was used.

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
