# Peer review of "Effective Bio-Slime Coating Technique for Concrete Surfaces under Sulfate Attack"

_materials, 2020, doi:10.3390/ma13071512_

Round 1

Reviewer 1 Report

In this paper the authors investigate an intriguing problem of improving concretes against sulfate attacks, using bacteria-based repair coating layers. The service life and sulfate diffusion coefficient were studied. The paper is well written and structured. I did not detect any specific mathematical or physical errors. However, I have several comments that the authors should consider when revising their manuscript:

  • I am missing a more detailed description of the chloride diffusion experiment and plots of the corresponding measured data from which chloride diffusivity was calculated. Without these the presented results concerning this issue cannot be assessed.
  • In Sec. 4.3 a concise explanation of how the evaluations of values of the service life and sulfate diffusivity was carried out is missing.
  • The diffusivities D1 and D2 in Eq. (2) do not seem to be time dependent, although this is explicitly stated in the text. A right parenthesis is missing in the second line of Eq. (2b).
  • Table 1 is spread across two pages and its content is rather limited.
  • I would rather use ‘1/2’ as a power instead of ‘0.5’; and ‘s’ instead of ‘sec’.

Author Response

We are very much thankful to you for reviewing our paper. The authors made the revision in the text of the paper based on the reviewer’s comments and suggestions. We hope our paper is much more suitable for publication in your prestige journal.

1. I am missing a more detailed description of the chloride diffusion experiment and plots of the corresponding measured data from which chloride diffusivity was calculated. Without these the presented results concerning this issue cannot be assessed.

-> As commented, we added detail description of the chloride diffusion experiment and calculation of diffusion in Sec. 4.1.

“In the NTBUILD 492 method, 10 % NaCl solution is applied for cathode, and 0.3 N NaOH solution is adapted for anode to accelerate penetration of chloride ion. Based on initial current and the criteria specified in the method, applied voltage and test period is determined. After the voltage is applied each test period, silver nitrate solution (0.1 M, AgNO3) is used as indicator and diffusion coefficient was calculated through Eq. (3) and Eq. (4).

2. In Sec. 4.3 a concise explanation of how the evaluations of values of the service life and sulfate diffusivity was carried out is missing.

-> As explained in 4.1, the diffusion coefficient of sulfate ion was hard to evaluate. From the previous researches [3], the ratio of sulfate diffusion to chloride diffusion was in the level of 38.5%~50%, so that 40% of diffusion ratio was assumed for the analysis. Eq (1) was used and input data was listed in Table 4. The results from Atkinson model were shown in Fig.5. The results from FDM analysis were listed in Fig.6~Fig.12 for simulations. 

3. The diffusivities D1 and D2 in Eq. (2) do not seem to be time dependent, although this is explicitly stated in the text. A right parenthesis is missing in the second line of Eq. (2b).

-> As commented, D1 and D2 were constants for diffusion at the reference time (28 days). In the previous model [a], the time effect was not considered, so that the developed model with time effect was used [21]. In the model [22], diffusion coefficient was controlled with time exponent-m and specific period (tc). The Equation was corrected as commented.

[22] Andrade, C; Diez. J.M.; Alonso, C. Mathematical modeling of a concrete surface “skin effect” on diffusion in chloride contaminated media. Adv. Cem. Based Mater. 1997, 6(2), 39-44.

4. Table 1 is spread across two pages and its content is rather limited.

-> As commented, we modified Table 1 and added Table 2.

5. I would rather use ‘1/2’ as a power instead of ‘0.5’; and ‘s’ instead of ‘sec’.

-> As commented, they were corrected.

Reviewer 2 Report

Dear Authors

There are my comments.

1. In the introduction part the description of mechanism of interaction between the bacteria and concrete should be placed.

2. In the section 2. Preparation of Bio-Slime Coating Layer there is the information “were mixed at the same mass ratio and used” what was the mass ratio? Table 1 is incomprehensible.

3. All information presented at Figure 1 and 2 should be described in detail in text.

4. In Figure 4 the column chart should be used.

5. Table 7 is incomprehensible.

Author Response

We are very much thankful to you for reviewing our paper. The authors made the revision in the text of the paper based on the reviewer’s comments and suggestions. We hope our paper is much more suitable for publication in your prestige journal.

1. In the introduction part the description of mechanism of interaction between the bacteria and concrete should be placed.

-> As commented, we added the following mechanisms for bacterial concrete regarding enhancement of concrete performance;

“Protective coating technology with fixed bacteria can be accepted as a new extension of biological self-healing for concrete structures exposed to extreme environments such as moisture, chemicals, salts and microbiological attacks[19]. It might be expected that a protective biological coating material has sustainable and biological resistances against extreme environments, unlike the conventional coating materials, such as epoxy, which lead to a gradual deterioration under chemical or microbiological attacks. The biological coating mortars were produced using expanded vermiculite, used as bacterial carriers; natural sand; and nearly neutral binder. As a block membrane against exogenous attack, the glycocalyx is introduced as a substance produced around bacterial cell during cultivate[13].” We also enhanced the related reference [19]

2. In the section 2. Preparation of Bio-Slime Coating Layer there is the information “were mixed at the same mass ratio and used” what was the mass ratio? Table 1 is incomprehensible.

-> As commented, we enhanced the explanation on the modified Table 1. 

3. All information presented at Figure 1 and 2 should be described in detail in text.

-> As commented, the detailed explanation was enhanced for Fig.1 and Fig. 2 was deleted.

For Fig.1:

“Fig. 1 shows the material preparation and the overview of the bio-slime coating layer. To promote a sustainable and protective biomimetic effect to the coating mortar, the bacterial glycocalyx is introduced as a block membrane. The glycocalyx, formulated during the cultivation of bacteria, is consisted of a sparse slime layer and a polymer skin capsule that hardly surrounds a cell. Although the primary chemical composition of glycocalyx includes rhamnose, mannose, galactose, glucosamine, phosphorus, and fatty acids. The compositional ratio varies depending on the composition of the bacterial strain and ingredients of the cultivate media."

4. In Figure 4 the column chart should be used.

-> As commented, the figure was improved with column chart.

5. Table 7 is incomprehensible.

-> The order numbering was wrong so that it was corrected and the related content was added in the Sec. 4.1. The previous Table 7 was a normal concrete mix proportions for the structure, not for coating, so that it was deleted.

“The target structure was RC box structure exposed to high concentration of sulfate, and the mix proportions were assumed to have 30% of GGBFS replacement ratio and the design strength of 35MPa, which was conventionally used for resisting chemical attack.”

Round 2

Reviewer 1 Report

The authors have incorporated all my suggestions, so I recommend the paper be published in the Materials.

Reviewer 2 Report

Dear Authors

I accept the correction you made